# Gut Microbiome Alterations, Mental Health, and Alcohol Consumption: Investigating the Gut–Brain Axis in Firefighters

**DOI:** 10.3390/microorganisms13030680

**Published:** 2025-03-18

**Authors:** Ji Youn Yoo, Anujit Sarkar, Hyo-Sook Song, Sunghwan Bang, Gyusik Shim, Cary Springer, Morgan E. O’Brien, Yoonhwa Shin, Songhyun Ju, Sunhee Han, Sung Soo Kim, Usha Menon, Tae Gyu Choi, Maureen E. Groer

**Affiliations:** 1College of Nursing, University of Tennessee, Knoxville, TN 37996, USA; asarkar7@utk.edu (A.S.); mgroer@utk.edu (M.E.G.); 2Department of Paramedicine, Bucheon University, Bucheon 14632, Republic of Korea; 20243001@woosuk.ac.kr (H.-S.S.); paramedic8@bc.ac.kr (S.B.); 3Department of Paramedicine, Korea Nazarene University, Cheonan 31172, Republic of Korea; sks9619@kornu.ac.kr; 4Research Computing Support, Office of Innovative Technologies, University of Tennessee, Knoxville, TN 37996, USA; springer@utk.edu; 5Department of Public Health, University of Tennessee, Knoxville, TN 37996, USA; sxg417@vols.utk.edu; 6Department of Biochemistry and Molecular Biology, School of Medicine, Kyung Hee University, Seoul 02447, Republic of Korea; jac03032@khu.ac.kr (Y.S.); thdgus8543@khu.ac.kr (S.J.); sunheehan@khu.ac.kr (S.H.); sgskim@khu.ac.kr (S.S.K.); 7College of Nursing, University of South Florida, Tampa, FL 33612, USA; umenon@usf.edu; 8Tampa General Hospital Cancer Institute, Tampa, FL 33606, USA; 9Department of Pathogenic Laboratory Research, Institute of Occupation and Environment, Korea Workers’ Compensation & Welfare Service, Incheon 21417, Republic of Korea; chtag@comwel.or.kr

**Keywords:** firefighters, alcohol consumption, stress, post-traumatic stress disorder (PTSD), gut microbiome, gut–brain axis

## Abstract

Firefighters across the world face higher risks of occupational hazards, such as exposure to chemicals, extreme heat, traumatic stressors, and intense physical demands, which can increase their vulnerability to a range of psychological and physiological difficulties. These challenges include the risk of developing chronic stress, depression, and post-traumatic stress disorder (PTSD), potentially leading to detrimental negative coping patterns such as alcohol abuse. The consequent health implications impact both short-term and long-term health and well-being. This study aimed to explore the relationship between mental health status, alcohol consumption patterns, and gut microbiome alterations in firefighters from two different regions—America and Korea. By investigating these relationships, we hope to gain insights into how repeated exposure to severe stressors impacts gut health. Healthy male firefighters (ages 21–50) and controls (matched sex, geography, and age) were recruited via flyers and snowball sampling in the United States and South Korea, resulting in 203 participants (102 firefighters and 101 controls). Firefighters reported significantly higher PTSD symptoms and depression and drank 2.3 times more alcohol than the control group. American firefighters reported more drinking than Koreans. There was a significant correlation between higher alcohol consumption and the likelihood of witnessing deaths by suicide. However, there were no correlations between alcohol consumption and PTSD symptom severity. There were associations between alcohol consumption patterns and aspects of the gut microbiome. This study highlights the mental health challenges faced by firefighters, including elevated rates of PTSD, depression, and alcohol consumption, with specific microbial imbalances linked to PTSD and alcohol use, emphasizing the role of the gut–brain axis.

## 1. Introduction

Firefighters perform vital services in communities worldwide by acting as first responders in a multitude of traumatic situations while minimizing property damage. In the process, they face the threat of injury, long 24-hour shifts, the realistic fear of death, and exposure to traumatic stressors such as the witnessing of suicides [1]. These challenges have led to early retirements, divorces, and the adoption of risky behaviors such as alcohol and drug misuse [2]. These risky behaviors, especially alcohol use, are often associated with stress and post-traumatic stress disorder (PTSD) as maladaptive coping mechanisms [3]. Research indicates that 52% of men and 28% of women with PTSD meet the criteria for alcohol abuse and dependence, compared to 25% of men and 11% of women without PTSD [4,5]. Additionally, studies have found that over 30% of firefighters experience alcohol abuse problems, which is about twice the rate observed in the general population [6]. There is evidence that heavy alcohol consumption is one of the negative coping mechanisms for dealing with post-traumatic stress in firefighters [7,8,9]. Despite growing awareness of the mental health challenges firefighters face, the complex relationship between psychological distress, alcohol use, and biological factors like the gut microbiome remains poorly understood.

Recently, the gut–brain axis, a bidirectional communication between the gut and brain, has been highlighted for its important role in mental health. An alteration of the gut microbiome, known as gut dysbiosis, has been associated with mental health problems, including PTSD, depression, and anxiety [10,11]. Similarly, the severity of alcohol use disorders has been associated with changes in the gut microbiome [12,13]. The transplantation of gut microbiota from severe alcohol consumers to mice caused changes in neuroinflammation and behavior in the mice [14]. These observations have led to a possible connection between the gut microbiome and alcohol consumption.

Despite growing recognition of the links between psychological stress, alcohol consumption, and gut microbiome health, limited research has examined how these factors interact in populations exposed to chronic stress, such as firefighters. Understanding the interrelationship between psychological distress, alcohol consumption, and gut microbiome changes may offer valuable insights into potential interventions that could mitigate stress and its associated health impacts.

We hypothesize that repeated exposure to severe stressors and traumatic events could lead to gut dysbiosis, independent of other key determinants of the gut microbiome, such as geographic, genetic, and dietary factors. To test this hypothesis, we selected firefighters from two different countries—America and Korea. This study aims to fill this gap by exploring the triangular relationship between the mental health status of firefighters, their alcohol consumption patterns, and gut microbiome alterations. Specifically, the objectives of the study are the following: (1) investigate the association between stress, anxiety, depression, and PTSD symptoms with alcohol consumption patterns; (2) examine the link between the severity of PTSD symptoms and gut microbiota composition and diversity; and (3) assess the relationship between alcohol consumption and gut microbiota composition and diversity while predicting potential microbial pathways in heavy alcohol drinkers compared to non- or light drinkers.

## 2. Materials and Methods

### 2.1. Participants Recruitment

Flyers and snowball sampling were used to recruit healthy male firefighters between 21 and 50 years of age, as well as a control group that was sex, geography, and age-matched in Tampa and St Petersburg, Florida; Knoxville and Johnson City in Tennessee; and Daejeon and Cheonan in South Korean from 2020 to 2023. A total of 253 participants were pre-screened; however, 50 participants did not provide biological samples or did not meet the study criteria and were therefore excluded. One firefighter’s stool sample quality could not be sequenced, and the sequencing data were excluded. A total of 203 participants were involved in this study, including 52 American firefighters, 50 American control individuals, 50 Korean firefighters, and 51 Korean controls. This study was approved by the University of South Florida Institutional Review Board, the University of Tennessee, Knoxville Institutional Review Board, and the Korean National Institution Bioethics Policy Committee.

#### 2.1.1. Inclusion Criteria

Firefighters: Active healthy male firefighters with a minimum of 5 years of active service; aged 21–50 years; non-smokers, non-tobacco users, or those who stopped using tobacco 5+ years ago.

Controls: Individuals who have never had a career or volunteer firefighter role, meeting the same demographic and health criteria as firefighters.

#### 2.1.2. Exclusion Criteria

Firefighters: Employed in 50% or higher administrative roles; diagnosed with chronic bowel disorders or inflammatory bowel disease; cancer history; diagnosed with human immunodeficiency virus, diabetes, autoimmune disorders, hepatitis A, hepatitis B, or hepatitis C; current user of pro/prebiotics; smoker; those who do not meet inclusion criteria. Controls: Meeting the same health criteria as firefighters.

### 2.2. Sample Collection and Data Analysis

For pre-screening questionnaires, online consent was obtained in REDCap, where all study survey data were stored. Self-administered questionnaires, as well as stool, urine, and hair samples, were collected from each participant after written informed consent was obtained. Approximately 30 mg of hair was collected from the scalp or arms (if an individual was bald). Stool samples were collected using the Easy Stool Collection Kit (Alpco, Salem, NH, USA). The participants were instructed to store all the samples immediately in a freezer until the collection day, which was less than 48 h from the collection point. A researcher visited fire stations or participants’ homes to collect the samples in an iced container to deliver to the lab.

### 2.3. Self-Questionnaires

The demographic and psychological health characteristics of the participants were measured using an investigator-developed questionnaire, and total scores were computed on the Generalized Anxiety Disorder-2 (GAD-2), Patient Health Questionnaire-2 (PHQ-2 Depression), Perceived Stress Scale (PSS), and the PTSD Checklist—Civilian Version (PCL-C). This instrument contains 17 items, each rated on a 5-point Likert scale (1 = Not at all to 5 = Extremely), reflecting the diagnostic criteria of PTSD. The total score is obtained by summing all item scores, resulting in a possible range from 17 to 85. Participants who achieved a total PCL-C score between 35 and 85 were identified as having significant PTSD symptoms and were provided with a referral for further psychological evaluation or intervention. Participants’ stress levels were assessed using the PSS, which consists of 10 items rated on a 5-point Likert scale (0 = Never to 4 = Very Often). Items 4, 5, 7, and 8 were reverse scored. After reverse scoring, the total score was calculated by summing the responses for all 10 items, yielding a possible score range of 0 to 40. Participants with a total PSS score between 27 and 40 were categorized as experiencing high perceived stress and were provided with a referral for further evaluation or support. If a participant had a score greater than or equal to 3 on either the GAD-2 or the PHQ-2, we considered them to have anxiety (GAD-2 score of 3 or above) and depression (PHQ-2 score of 3 or above), and we provided a referral. Furthermore, when we analyzed the data, we grouped participants based on whether their total GAD-2 or PHQ-2 score was below or above three. Each score was analyzed and compared using frequencies and descriptive statistics (IBM SPSS; Version 28.0 Armonk, NY, USA: IBM Corp.).

### 2.4. Alcohol Consumption

Alcohol consumption was measured according to the guidelines provided by the National Institute on Alcohol Abuse and Alcoholism (NIAAA). One standard drink (or one alcoholic drink equivalent) was defined as any beverage containing approximately 0.6 fluid ounces or 14 g of pure alcohol.

The following beverages were considered as examples of a standard drink:12 ounces of regular beer (approximately 5% alcohol by volume);5 ounces of wine (approximately 12% alcohol by volume);1.5 ounces of distilled spirits (approximately 40% alcohol by volume).

The frequency and amount of alcohol consumption was assessed. Excessive drinking was defined as consuming more than 12 ounces of alcohol per day.

### 2.5. Microbial Diversity, Composition Profiling, and Data Analysis

Stool bacterial DNA was extracted using the QIAamp DNA Stool Mini Kit (Qiagen, Germantown, MD, USA). We purified DNA for PCR amplification. The V4 region (primers 515F–806R) of the 16S rRNA gene was amplified and sequenced using the Illumina MiSeq platform to generate ~100,000 and 250 bp paired-end reads per sample. The raw sequencing data were processed using the standard DADA2 pipeline in R (Version 1.26.0) [15]. Briefly, the samples containing less than 2000 reads were discarded. The forward reads were trimmed by 20 bases, while the reverse sequences were trimmed by 50 bases to discard low-quality sequences. The phiX genome and reads with ambiguous bases were also discarded. The DADA2 algorithm was utilized separately to identify the Amplicon Sequence Variants (ASVs) in the study. In the next step, the ASVs were merged, and chimeras were identified and discarded. The filtered ASVs were classified using the Silva v132 database. Finally, the distribution of each ASV among the samples was calculated, and their relative abundances were determined.

The differences in the microbiome composition between the firefighters and the non-firefighters were determined by the Microbiome Multivariable Association with Linear Models (MaAsLin2) [16]. It utilizes linear modeling to identify associations between microbial features and metadata variables while correcting for provided covariates. In this task, the cohort was divided into those with PTSD and the controls. Similarly, the cohort was divided based on alcohol composition to identify the ASVs that significantly differed between the two groups. In the next step, the target was to identify the functional potential of the microbiome from predicted metagenomes. Consequently, the ASV abundance table for all samples and the ASV sequences obtained from the DADA2 pipeline were utilized to determine the function potential employing MicFunPred [17]. This computational tool can predict metagenomes from the 16S rRNA sequences in terms of Kyoto Encyclopedia of Genes and Genomes (KEGG) orthologs (KOs), Clusters of Orthologous Genes (COGs), Carbohydrate-Active enZYmes (CAZYs), Protein family database (Pfams), KEGG pathway abundances, etc., using MinPath [18].

As several of these predicted functional entities were quite rare and were not expected to have a considerable biological significance, only those that were present in at least 1% in one of the samples (abundance) and those present in at least 10% of all the samples (persistence) were retained for statistical analyses. To identify the differences between the two groups (FF vs. non-FF), ALDEx2 [19] was utilized, which corrects for the compositional nature of the microbiome data. Briefly, it utilizes centered log ratio (clr) transformation to handle the compositional nature of the microbiome data. All feature abundance values are expressed relative to the geometric mean abundance of all features in a sample. ALDEx2 was utilized to compare the abundances between the two groups (FF vs. non-FF) by employing Welch’s *t*-test, and the *p*-values were corrected by Benjamini–Hochberg false discovery rate (FDR) correction.

## 3. Results

### 3.1. Participant Recruitment

A total of 203 participants participated in the study, 102 firefighters and 101 controls, from America and South Korea. The study sample contained 50 South Korean firefighters, 52 American firefighters, 51 South Korean controls, and 50 American controls. The age range of the participants was 21–50 years old, with the majority of the participants being 31–40 years old (54% of South Korean firefighters, 46.2% of American firefighters, 45.1% of South Korean controls, and 36% of American controls). The South Korean participants entirely identified as Asian (100%), while the American participants primarily identified as white (92.3% of firefighters and 66% of non-firefighters). The majority of participants had a college degree or higher, with non-firefighters’ top occupations being researcher for American controls (24%) and education for South Korean controls (25.5%). Firefighters had a range of years in service; the majority of South Korean firefighters served for 5–10 years (64%), while American firefighters had a majority serving 5–10 years (40.4%) and more than 10 years (36.5%) (Table 1).

### 3.2. Post-Traumatic Stress Disorder (PTSD) Symptoms, Perceived Stress Scale (PSS), Anxiety, and Depression

Our analysis indicates that there was a significant and clinically meaningful difference in PTSD symptoms scores between all firefighters (American and Korean) and the control groups (American and Korean). The respective control groups reported a mean score of 25.24 (SD = 11.71), while the firefighter group’s mean score was slightly higher at 29.12 (SD = 10.6); this difference was statistically significant (*p* = 0.008) (Figure 1a). This trend of elevated PTSD symptoms scores was consistent across geographical locations, with both American and Korean firefighters reporting higher PTSD symptoms than their control groups (Figure 1b). We determined the PCL-C cutoff score for our population as 30. Other investigators have also evaluated the validity of the PCL-C and reported optimal cutoff scores. Walker et al. [20] found that among women attending a health maintenance organization (HMO) in Seattle, the optimal PCL-C cutoff score was 30, with a sensitivity of 82% and specificity of 76%. Similarly, Andrykowski et al. [21] reported that in a study of women with breast cancer, a PCL-C cutoff score of 30 yielded a sensitivity of 100% and specificity of 83%.

The two-way ANOVA revealed a significant interaction between geographic regions and occupation on PSS levels. There were no significant main effects of geographical regions or occupation (firefighters vs. non-firefighters); however, the interaction indicates that the PSS levels in firefighters and control participants differ based on their geographical regions. The control groups between Koreans (mean = 14.49, SE = 0.799) and Americans (mean = 11.38, SE = 0.807) showed significantly different PSS levels (*p* = 0.007), while the firefighter groups did not (American mean = 14.21, SE = 0.791, and Korean mean = 14.49, SE = 0.807) (*p* = 0.601). American firefighters reported significantly higher levels of perceived stress compared to the American control group (*p* = 0.013). However, this finding was not replicated in the Korean subgroups, where no significant difference was detected between Korean firefighters and their respective control group (*p* = 0.444). Interestingly, the Korean control group exhibited the highest levels of perceived stress among all groups (Figure 1c,d).

A multivariate logistic regression analysis was conducted to examine the impact of geographical region and occupation on the incidence of depression. The overall model was statistically significant (*p* = 0.036). When examining the individual predictors, geographical region was not a significant factor (*p* = 0.5), suggesting that there were no significant differences in depression incidence based on region. However, occupation was a significant predictor of depression (*p* = 0.013), indicating that occupation plays a critical role in predicting the likelihood of depression. Firefighters were found to have a twofold higher incidence of depression compared to the control group, with an odds ratio of Exp(B) = 2.04. The 95% confidence interval for this odds ratio ranged from 1.161 to 3.586. There were correlations among levels of PSS, PTSD symptoms, depression, and anxiety, indicating that these psychological and behavioral variables are interconnected (Table 2).

### 3.3. Alcohol Consumption Patterns

Our results show that firefighters (both American and Korean) consumed alcohol at a rate 2.3 times higher than the control group (*p* = 0.012). The proportion of non-drinkers within the control group stood at 38.6%, compared to a lower prevalence of 28.4% among firefighters. In contrast, excessive drinking was more common among firefighters (24.5%) compared to the control group (11.9%) (*p* = 0.048) (Figure 2a). There was also a significant difference between American and Korean participants: 51% of Koreans were non-drinkers, while only 17% of Americans were non-drinkers (*p* < 0.001) (Figure 2b).

We asked both firefighter groups (*n* = 102) whether they had ever observed a suicide attempt (80.4%) or suicide death (74.5%) during their work and examined the association between these observations and alcohol consumption. There was no significant association between observing suicide attempts and alcohol consumption categories (*p* = 0.725); however, there was a significant association between observing suicide deaths and alcohol consumption among firefighters (*p* = 0.003). Specifically, individuals who observed suicide deaths were more likely to become moderate (51.3%) or excessive drinkers (28.9%) compared to non-drinkers (19.7%) (Figure 2c).

### 3.4. Gut Microbiota Diversity and Composition

Firefighters’ gut microbiota diversity metrics were not significantly different from the control groups (Figure 3a). However, the geographic subgroups, specifically American versus Korean, displayed distinct differences in gut microbiota diversity (Figure 3b).

### 3.5. Gut Microbiota Diversity and Composition with PTSD Symptoms

We used a cutoff score of 30 on the PCL-C, which indicates experiencing a considerable number of PTSD symptoms, to distinguish individuals with PTSD symptoms. The gut microbiome alpha diversity was slightly higher in individuals without PTSD symptoms compared to those with PTSD symptoms across all groups, meaning the groups were not stratified by geography or occupation. However, the difference was not statistically significant (Figure 4a, Shannon: *p* = 0.31, Simpson: *p* = 0.2). A similar pattern was observed among firefighters, where those with higher PTSD symptom levels had slightly lower alpha diversity than those without PTSD. However, the statistical *p*-values were not significant, which may be due to the small sample size (Figure 4b, Shannon: *p* = 0.34, Simpson: *p* = 0.42). We found *Alistipes putredinis* to be more prevalent in individuals without PTSD symptoms (n = 124) compared to those with PTSD symptoms (n = 65) across all four groups. In contrast, *Veillonella torques* exhibited the opposite trend, being more prevalent in individuals with PTSD symptoms (Figure 4c,d).

### 3.6. Microbiota Composition and Diversity in the Pattern of Alcohol Consumption Frequency

The analysis discovered distinct differences in the microbiota composition between individuals based on their alcohol consumption frequency. Key bacterial taxa, including members of the Lachnospiraceae family and *Bacteroides_s* genus, were differentially abundant across alcohol consumption patterns. Among the Lachnospiraceae family, significant differences were observed in several Amplicon Sequencing Variants (ASVs). Specifically, *Lachnospiraceae_xylanophilum_group*, *Lachnospiraceae_UCG*, *Lachnospiraceae_s*, and *Bacteroides_s* showed increased abundance in individuals with higher alcohol consumption frequency, whereas *Bifidobacterium_s* was more prevalent in those with non- or minimal alcohol intake (Figure 5a–e).

### 3.7. Microbiota Composition and Diversity in the Pattern of Alcohol Consumption Amount

We also analyzed the gut microbiota based on the amount of alcohol consumed, and the results showed a similar pattern to that observed with alcohol consumption frequency. For example, several taxa, particularly those belonging to the Lachnospiraceae family, including *Lachnospiraceae Tuzzerella*, *Lachnospiraceae_xylanophilum_group*, and *Lachnospiraceae_UCG*, exhibited variable abundance patterns associated with higher alcohol consumption amounts, while *Bifidobacterium bifidum* showed reduced abundance in high alcohol consumption groups (Figure 5f,g).

### 3.8. Microbial Function Prediction Using KEGG Orthologs (KOs)

As described previously, various classes of functionally relevant molecules contributed by the gut microbiome were predicted. The differentially abundant KEGG orthologs (KOs) between the firefighter groups and the control groups are shown in Figure 6. The 2-Dehydro-3-deoxyphosphogluconate aldolase (EDA) was predicted to be significantly higher in the firefighter groups than the control groups (*p* = 0.005) after correction for multiple testing. EDA is an enzyme which is involved in the Entner–Doudoroff (ED) pathway, which is a central metabolic route in certain bacteria and archaea.

In contrast, the abundant Dipeptidases KOs, Sortase A (SrtA), and DNA segregation ATPase FtsK were predicted to be significantly higher in the control groups than the firefighter groups (*p* = 0.008, *p* = 0.009, *p* = 0.016). Dipeptidases are enzymes that belong to the class of peptidases (proteases) specifically responsible for hydrolyzing dipeptides into their constituent amino acids. These enzymes play a key role in protein digestion and metabolism, as well as regulating insulin levels in the blood. SrtA plays a key role in anchoring surface proteins to the cell wall in Gram-positive bacteria, which contributes significantly to bacterial pathogenicity and biofilm formation. DNA segregation ATPase FtsK is a large ATPase that uses energy from ATP hydrolysis to mediate the movement of DNA, ensuring proper chromosome segregation during the bacterial cell cycle (Figure 6).

Upon comparing the Carbohydrate-Active enZYmes (CAZYs), our results also show that firefighters had predicted lower levels of Glycoside Hydrolase Family 25, Glycosyl Transferase Family 51, Carbohydrate-Binding Module Family 13, and Glycosyl Transferase Family 35 compared to the control group. These microbial enzyme families play crucial roles in carbohydrate metabolism and digestion.

## 4. Discussion

We hypothesized that repeated exposure to severe stressors and traumatic events could lead to gut dysbiosis independently of other key determinants of the gut microbiome, such as geographic and genetic factors. To test this, we selected firefighters from two different countries—America and Korea. Our data demonstrated a connection between exposure to traumatic events, alcohol consumption, and alterations in the gut microbiome. While our analysis revealed significant differences in gut microbiome diversity between American and Korean participants, we did not observe differences in diversity between firefighters and non-firefighters, contrary to our initial hypothesis. However, our study provided meaningful evidence showing that gut dysbiosis was associated with both severe PTSD symptoms and heavy alcohol consumption patterns. This suggests that, despite similar overall microbial diversity, specific microbial imbalances linked to PTSD symptoms levels and higher alcohol use may contribute to disruptions in gut health. We focused on the correlation between PTSD and gut microbiota rather than anxiety and depression due to the high prevalence and severity of PTSD among firefighters, who frequently encounter traumatic events such as suicides and severe injuries. While anxiety and depression are also common, PTSD is particularly relevant to their occupational exposure. Additionally, although anxiety and depression have been extensively studied in relation to gut microbiota, PTSD remains underexplored. Given the heightened risk of PTSD in firefighters, our study aims to fill this gap by examining its association with microbial diversity, composition, and functional pathways. We acknowledge the significance of anxiety and depression in mental health research and will explore their relationships with gut microbiota in future studies.

### 4.1. PTSD Symptoms and Other Mental Health Outcomes

Our study shows that firefighters reported significantly higher PTSD symptoms compared to the control groups, which supports the previous literature. Kessler (2005) and other studies revealed that firefighters were two to three times more likely to develop PTSD compared to the general population [22,23]. Over the past decade, increasing attention has been directed toward the mental health of firefighters that is driven by the growing evidence of elevated rates of psychiatric disorders, including depression, anxiety, and alcohol abuse among firefighters [24]. Our study is the first to demonstrate that the trend of PTSD, depression, and anxiety risk in firefighters remains consistent across geographical regions, highlighting the global nature of occupational stress experienced by firefighters.

Our study showed mixed findings when comparing the perceived stress scale (PSS) levels between firefighters and the control groups. Broadly, there were no significant differences between the two groups. However, further subgroup analysis shows that American firefighters reported significantly higher PSS levels compared to their American control group. This discrepancy might be attributed to unique stressors faced by American firefighters, such as a higher frequency of traumatic events, differing work conditions, or perhaps societal expectations around managing occupational stress [25,26].

Interestingly, this pattern was not observed in the Korean subgroups, where no significant difference in PSS was found between Korean firefighters and their control counterparts. Remarkably, the Korean control group reported the highest PSS levels across all four groups. Several factors may explain this unexpected finding, such as cultural and societal factors. For example, cultural differences in stress perception and reporting likely play a role, particularly in East Asian societies like Korea, where seeking treatment for stress is often stigmatized, with going to the hospital for stress-related issues being viewed negatively. This cultural stigma may cause individuals to avoid seeking help and underreport their stress levels, especially in professional settings like firefighting, where resilience is highly valued [27]. Conversely, the higher PSS levels in the Korean control group might reflect broader societal stressors prevalent in the general population, such as economic pressures and social expectations [28]. These factors may have elevated baseline stress levels in the control group, potentially diminishing observable differences between firefighters and controls in the Korean population. Future studies should consider these cultural and societal influences when evaluating stress and its psychological impacts across different populations.

Our study also found a statistically significant association between the occupation of firefighting and the prevalence of depression, with firefighters experiencing a twofold higher incidence of depression (52.9%) compared to the control group. This finding aligns with the existing literature that highlights the elevated mental health risks faced by individuals in high-stress occupations like firefighting. Frequent exposure to traumatic events, the demanding nature of the job, and the pressure to remain resilient in the face of adversity may contribute to this increased prevalence of depression [29].

Although the association between being a firefighter and anxiety was not statistically significant, firefighters still reported a 71.3% higher incidence of anxiety compared to the control group, indicating a potential relationship between firefighting and heightened anxiety levels. This finding suggests that anxiety, like depression, may be more prevalent among firefighters, likely driven by occupational stress, irregular work schedules, and exposure to life-threatening situations. Our results, alongside existing research, highlight the common co-occurrence of depression and anxiety with PTSD, reflecting the interconnected nature of these psychological and behavioral conditions [30,31,32]. The positive relationships among stress, PTSD symptoms, depression, and anxiety in our study reinforce the concept of a cumulative psychological burden, where chronic exposure to stress and trauma exacerbates mental health issues.

This pattern of co-occurring conditions is well documented in trauma-related fields [29,30,32,33], underscoring the need for comprehensive mental health support for firefighters. Our study also found a strong co-occurrence of symptoms, with 86.2% of individuals with PTSD also exhibiting symptoms of anxiety and/or depression.

Notably, there were no statistically significant differences between subgroups (American and Korean) in terms of depression and anxiety, suggesting that the increased prevalence of these conditions among firefighters is consistent across geographical regions. This consistency underscores the universal nature of the mental health challenges faced by firefighters, regardless of cultural or regional differences. However, cultural factors may influence how these mental health conditions manifest or are reported, as seen in our analysis of perceived stress levels. Therefore, while interventions should be globally informed, they must also be culturally tailored to ensure their effectiveness in different contexts.

### 4.2. Alcohol Consumption Patterns

Our study highlights significant differences in alcohol consumption patterns between firefighters and the control group, indicating that firefighters are at a greater risk for excessive alcohol use. Firefighters consumed alcohol at a rate 2.3 times higher than the control group, with a lower proportion of non-drinkers (28.4%) compared to the control group (38.6%). Additionally, excessive drinking was more prevalent among firefighters (24.5%) compared to controls. Although our study shows the absence of a direct correlation between alcohol consumption and PTSD symptoms, there was an association with witnessing traumatic events such as suicide deaths. This may be explained by the heterogeneity of PTSD symptoms. PTSD can manifest in different ways, and individuals may vary in their coping mechanisms [32]. Some firefighters with PTSD may not rely on alcohol to manage their symptoms, instead using other coping strategies such as avoidance, social withdrawal, or professional counseling [34]. The variability in how individuals respond to PTSD may explain why alcohol consumption is not universally linked to PTSD symptom levels across all firefighters. However, higher alcohol consumption in firefighters still occurs and can lead to health problems such as gut dysbiosis. Gut dysbiosis has been associated with many diseases, including cardiovascular and other metabolic conditions [35,36,37,38].

### 4.3. Gut Microbiota Diversity and Composition with PTSD Symptoms

Our study showed important differences in the gut microbiome between individuals with and without PTSD symptoms, highlighting a potential biological link between gut microbiota composition and mental health outcomes. Specifically, we found that alpha diversity, a measure of the variety and abundance of microbial species within the intestine, was greater in individuals without PTSD symptoms compared to those with PTSD among all the groups.

When focusing on the firefighter population, who reported higher levels of PTSD symptoms compared to control groups, we observed the same pattern: firefighters without PTSD symptoms had greater gut microbiota diversity than those with PTSD. This finding reinforces the growing body of evidence suggesting that higher gut microbial diversity may play a protective role in promoting resilience against the development of PTSD symptoms [38]. A diverse gut microbiome could be influencing the body’s stress responses and mental health by regulating inflammation, neurotransmitter production, and the gut–brain axis [39]. The gut–brain axis theory offers critical insights into how traumatic stress, such as PTSD, disrupts biological systems. This bidirectional communication network links the gut and brain, where prolonged stress-induced dysregulation of stress axes may lead to gut dysbiosis and systemic inflammation. While acute stress-induced cortisol release suppresses proinflammatory cytokines, persistent stress compromises immune function, leading to the redistribution of NK cells, impaired T and B cell responses, and elevated proinflammatory cytokines. Stress also reduces gut microbial diversity and weakens gut barrier integrity, further exacerbating dysbiosis—a microbial imbalance linked to anxiety, depression, PTSD, metabolic disorders, and cancer [40,41].

These results further underscore the potential of gut microbiome-based treatments in modulating stress responses and enhancing overall psychological resilience.

In terms of specific bacterial taxa, *Alistipes putredinis* was more prevalent in individuals without PTSD symptoms (n = 124), while *Veillonella torques* was more abundant in individuals with PTSD (n = 65) among all the groups. The higher prevalence of *Alistipes putredinis* in those without PTSD may indicate a beneficial role of this bacterium in maintaining well-being, as prior research has suggested that this genus may be linked to stress resilience in animal models [42] and longevity in humans [43]. *Alistipes putredinis,* a commensal bacterium found in the human gastrointestinal tract, contributes to the digestion of complex carbohydrates and the fermentation of dietary fibers, producing short-chain fatty acids (SCFAs) such as butyrate [44]. SCFAs are crucial for gut health as they help maintain the intestinal barrier, reduce inflammation, and provide energy to colon cells [45,46]. These functions could potentially underlie the protective effect of *Alistipes putredinis* in individuals without PTSD, as gut health and psychological health are increasingly understood to be interconnected through the gut–brain axis. Furthermore, a study demonstrated that *Alistipes putredinis* was more abundant in Japanese centenarians and their family members compared to Japanese young and older individuals, identifying it as a potential marker of successful aging [47]. However, it is important to note that recent studies have also associated *Alistipes putredinis* with colorectal cancer [48]. This dual role suggests that while this bacterium may offer benefits in stress resilience and gut health, its effects may be context-dependent, potentially contributing to disease under certain conditions. This complexity underscores the importance of further research to explore the mechanisms by which *Alistipes putredinis* could influence both mental and physical health outcomes, including its role in both promoting resilience and, potentially, in contributing to disease pathogenesis.

Conversely, the greater abundance of *Veillonella torques* in individuals with PTSD could reflect a microbial signature associated with stress-related disorders, as species within this genus, such as *Veillonella parvula*, have been implicated in inflammatory bowel disease [49] and other negative health outcomes [50]. However, the functions of *Veillonella torques* are not yet fully understood.

These findings further support the growing recognition of the gut–brain axis—the bidirectional communication between the gut microbiota and the central nervous system—as a crucial factor in mental health [51]. Reduced microbial diversity and shifts in specific bacterial taxa may contribute to the onset or exacerbation of PTSD symptoms, highlighting the potential of gut microbiota composition as a target for future interventions aimed at improving mental health outcomes in high-risk populations, such as firefighters.

### 4.4. Gut Microbiota Composition and Diversity in the Pattern of Alcohol Consumption Frequency and Amount

Our study results highlight alcohol consumption frequency and amount impact on gut microbial communities. Key bacterial taxa, particularly members of the *Lachnospiraceae* family and genus level of *Bacteroides*, displayed differential abundance across alcohol consumption patterns. Specifically, taxa such as the *Lachnospiraceae_xylanophilum_group*, *Lachnospiraceae_UCG*, *Lachnospiraceae_s*, and *Bacteroides_s* were more prevalent among individuals with higher alcohol consumption frequency. Likewise, when examining alcohol consumption amounts, similar taxa from the *Lachnospiraceae* family, including *Lachnospiraceae Tuzzerella*, *Lachnospiraceae_xylanophilum_group*, and *Lachnospiraceae_UCG*, showed increased abundance in individuals with greater alcohol intake. Other studies have also shown an enrichment of *Lachnospiraceae* in individuals with heavy alcohol consumption, supporting our findings. This family of bacteria has been consistently associated with altered gut microbiota profiles in the context of alcohol use. For example, research in human cohorts has demonstrated similar increases in *Lachnospiraceae* abundance among heavy drinkers, suggesting that alcohol consumption may create a favorable environment for the proliferation of these taxa. Additionally, a study using a mouse model demonstrated the distinct role of *Lachnospiraceae* in the gut under conditions of alcohol exposure, further indicating that this bacterial family is responsive to alcohol-induced changes in the gut environment [52,53] Conversely, beneficial bacteria such as *Bifidobacterium_s* and *Bifidobacterium bifidum* were found to be less prevalent among individuals with higher alcohol consumption, suggesting that excessive alcohol intake may have negative effects on beneficial gut bacteria, leading to potential dysbiosis [54]. This reduced abundance of *Bifidobacterium* species, known for their roles in maintaining gut health and immune function, emphasizes the perception that alcohol disrupts the gut microbiome, potentially contributing to adverse health outcomes. Alcohol consumption can alter the abundance of beneficial bacteria, such as *Bifidobacterium bifidum*, through mechanisms primarily related to increased gut permeability and endotoxemia [55,56].

These findings emphasize the consistent influence of both the frequency and amount of alcohol consumption on gut microbiota composition, particularly the increase in specific taxa associated with higher alcohol intake and the reduction of beneficial bacteria.

### 4.5. Microbial Functional Prediction Using the KEGG Orthologs (KOs)

Our analysis of KOs between firefighters and control groups revealed significant differences in the abundance of several key microbial enzymes and pathways, providing insights into potential functional shifts in the gut microbiome of firefighters. Specifically, 2-Dehydro-3-deoxyphosphogluconate aldolase (EDA), an enzyme involved in the Entner–Doudoroff (ED) pathway, was significantly predicted to be more abundant in the firefighter group compared to the controls. The ED pathway, an alternative to glycolysis, efficiently breaks down glucose into pyruvate and glyceraldehyde 3-phosphate (G3P) with fewer enzymatic steps and is often used in nutrient-poor or semi-aerobic environments [57]. This pathway is crucial for energy metabolism in various microbes [58]. The elevated levels of EDA in firefighters may indicate a microbiome shift toward enhanced glucose metabolism, potentially reflecting stress-induced changes in nutrient processing. However, current evidence is insufficient to directly link the ED pathway in gut bacteria to specific health outcomes in humans.

In contrast, the control group exhibited prediction of higher levels of several important microbial enzymes, such as dipeptidases, Sortase A, and DNA segregation ATPase FtsK. Dipeptidases, crucial for hydrolyzing dipeptides into amino acids, play a significant role in protein digestion and metabolism, particularly in the small intestine, where they assist in the final stages of digestion and insulin regulation [58,59]. Lower dipeptidase abundance in firefighters could result in incomplete protein digestion, potentially leading to amino acid malabsorption and nutritional deficiencies. This enzymatic imbalance might also contribute to intestinal inflammation [60], which could negatively affect overall metabolic health and well-being.

Sortase A, predominantly found in Gram-positive bacteria, anchors surface proteins to the bacterial cell wall [61] and is well known as a mediator that plays a key role in the pathogenesis of *S. aureus* infections [62]. However, Sortase A is also found not only in pathogenic bacteria but also in normal gut microbiota. As a result, a lower abundance of Sortase A in firefighters may suggest a reduced presence of certain beneficial Gram-positive bacteria which are crucial for maintaining gut barrier function, modulating immune responses, and producing SCFAs [63]. The reduction in these bacteria could compromise gut integrity and immune function, potentially increasing the risk of gut inflammation or gastrointestinal issues, such as leaky gut syndrome [64].

Filament temperature-sensitive mutant K (FtsK), an essential ATPase involved in bacterial chromosome segregation and cell division [65], was also found in lower abundance in firefighters. Although research on FtsK and its relevance to human health is limited, its family member, Filamenting temperature-sensitive mutant Z (FtsZ), is a well-studied bacterial protein critical to cell division. Targeting FtsZ has been a promising approach in developing antibiotics that inhibit bacterial cell division by preventing proper DNA distribution, which is crucial for bacterial survival [66]. A reduction in FtsK-expressing bacteria may indicate shifts in the microbial community dynamics in the gut microbiomes of firefighters, potentially impairing gut health and immune function and increasing vulnerability to trauma-related or metabolic conditions.

Further analysis also revealed that firefighters had significantly lower levels of several glycoside hydrolase families and glycosyl transferase families, specifically Glycoside Hydrolase Family 25, Glycosyl Transferase Family 51, Carbohydrate-Binding Module Family 13, and Glycosyl Transferase Family 35. These enzyme families are essential for carbohydrate metabolism and digestion, with glycoside hydrolases breaking down complex carbohydrates into simpler sugars and glycosyl transferases catalyzing the transfer of sugar moieties to various acceptors [67,68]. The reduction in these enzymes in firefighters suggests impairments in carbohydrate metabolism, potentially leading to suboptimal energy production and nutrient utilization [69].

## 5. Conclusions

This study highlights the significant mental health challenges faced by firefighters, including elevated rates of PTSD, depression, anxiety, and alcohol consumption. While overall gut microbiome diversity did not differ between firefighters and controls, specific microbial imbalances were linked to PTSD and alcohol use, suggesting a role for the gut microbiome in stress resilience and mental health. The associations of *Alistipes putredinis* with PTSD and the impact of alcohol consumption on bacterial taxa like *Lachnospiraceae* and *Bifidobacterium* underscore the importance of the gut–brain axis in firefighters’ well-being. Additionally, shifts in microbial pathways related to glucose metabolism and protein digestion may reflect stress-induced changes in the microbiome.

## 6. Study Limitation

This study has several limitations. First, the sample size may limit the generalizability of our findings. Additionally, dietary intake and BMI data were not collected, despite their known influence on gut microbiome composition. The absence of these factors may have introduced potential confounding effects, making it difficult to fully interpret the observed microbiome differences. Future research should incorporate dietary and BMI data to provide a more comprehensive understanding of the gut microbiome’s role in firefighter mental health. Moreover, we acknowledge that the observed differences in gut microbiome diversity between American and Korean participants could be attributed to significant intra- and interpersonal variability of the gut microbiota. This variability is influenced by multiple factors, including lifestyle and environmental exposures, which were not fully accounted for in this study. Future studies should consider these variables to better understand the complex interactions between gut microbiota and mental health outcomes in diverse populations.

## Figures and Tables

**Figure 1 microorganisms-13-00680-f001:**
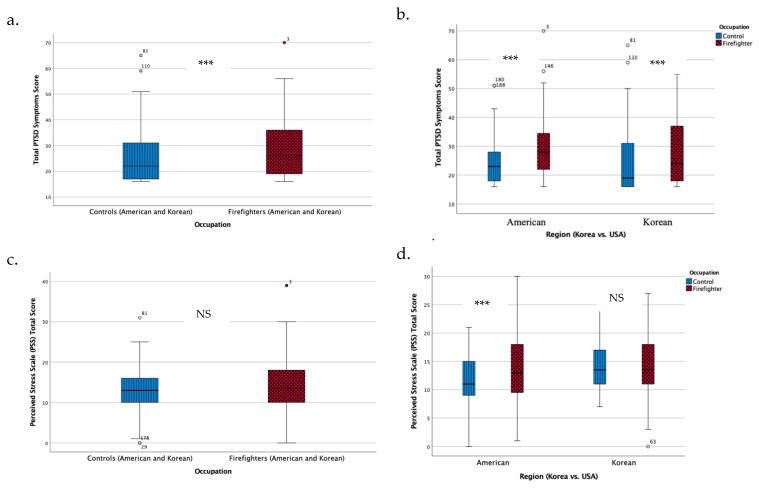
Associations between post-traumatic stress symptoms (PTSD Symptoms) and perceived stress scale (PSS) levels by occupation and geographical location: (**a**) PTSD symptoms across different occupational groups, (**b**) PTSD symptoms by geographical group, (**c**) PSS across occupational groups, and (**d**) PSS by geographical group. The notations *** corresponds to *p*-values of ≤0.001, respectively. NS represents not significant.

**Figure 2 microorganisms-13-00680-f002:**
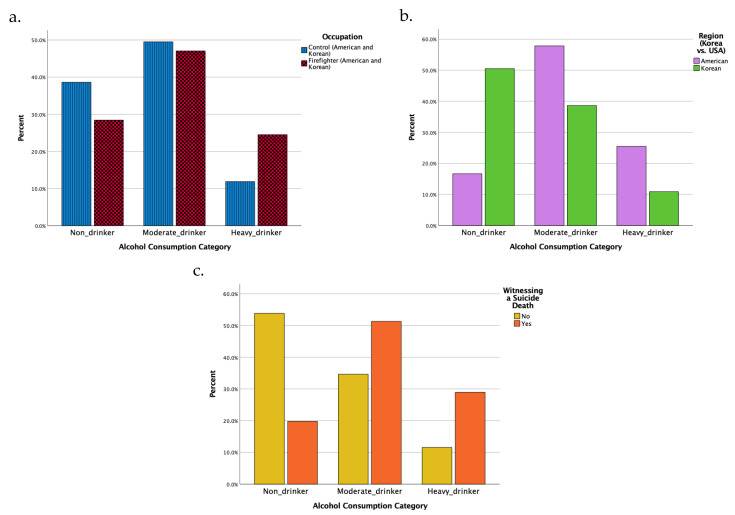
Alcohol consumption patterns and influencing factors among firefighters and controls: (**a**) Comparison across different occupational groups. (**b**) Comparison across different geographical groups. (**c**) Experience of witnessing a suicide death during work.

**Figure 3 microorganisms-13-00680-f003:**
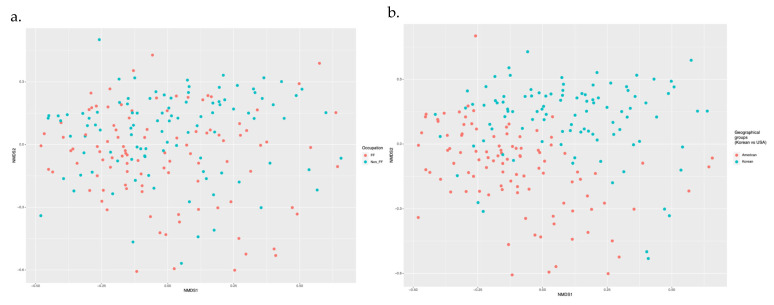
Gut microbiota diversity: (**a**) Comparison across different occupational groups. (**b**) Comparison across different geographical groups.

**Figure 4 microorganisms-13-00680-f004:**
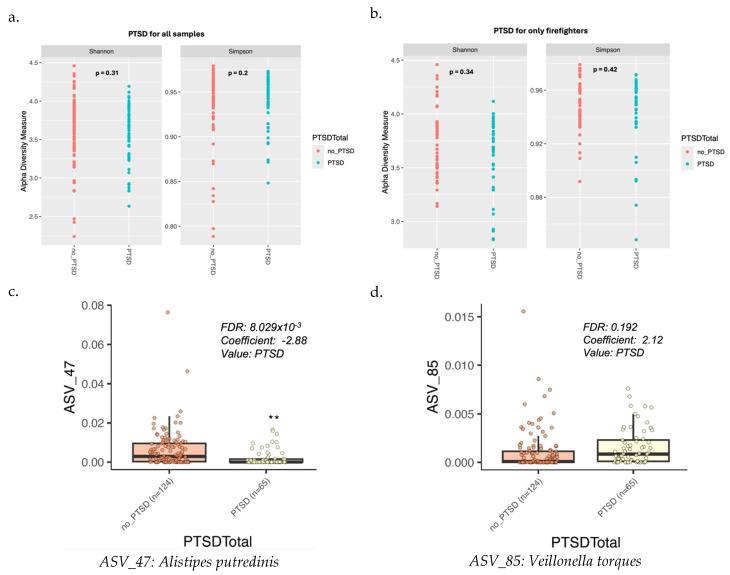
Gut microbiome diversity and bacterial composition by PTSD symptoms status: (**a**) Alpha diversity comparison between individuals with PCL-C total score below 29 (no PTSD symptoms) vs. above 30 (some PTSD symptoms), (**b**) alpha diversity among firefighters by PTSD symptoms status, (**c**) *Alistipes putredinis* (ASV 47) prevalence by PTSD symptoms status, and (**d**) *Veillonella torques* (ASV 85) prevalence by PTSD symptoms status.

**Figure 5 microorganisms-13-00680-f005:**
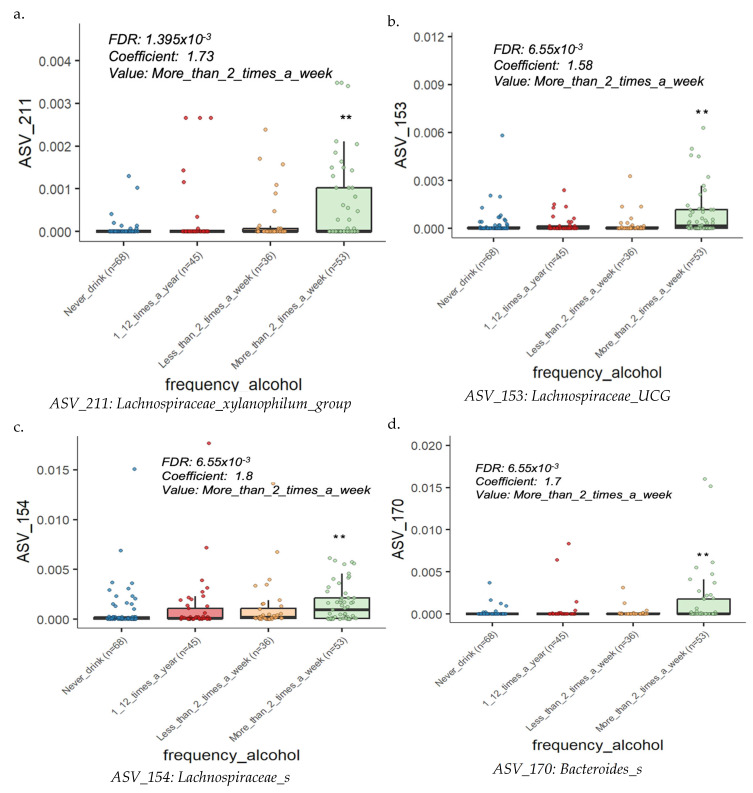
Microbiome composition and diversity in the pattern of alcohol consumption frequency and amount. (**a**) *Lachnospiraceae*_*xylanophilum_group* (ASV 211), (**b**) *Lachnospiraceae_UCG* (ASV 153), (**c**) *Lachnospiraceae_s* (ASV 154), (**d**) Bacteroides_s (ASV 170), (**e**) *Bifidobacterium_s* (ASV 10), (**f**) *Lachnospiraceae Tuzzerella* (ASV 174), (**g**) *Lachnospiraceae_xylanophilum_group* (ASV 211), (**h**) *Lachnospiraceae_UCG* (ASV 153), (**i**) *Bifidobacterium bifidum* (ASV 88). The notations * and ** corresponds to *p*-values of ≤0.05 and ≤0.01, respectively.

**Figure 6 microorganisms-13-00680-f006:**
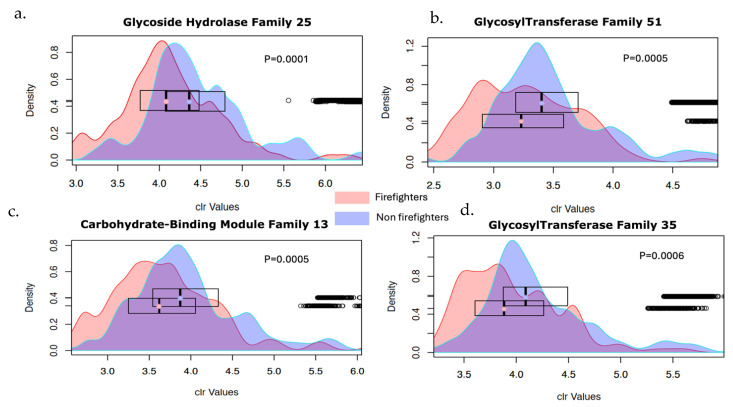
Density plots of differentially abundant KEGG orthologs (KOs) between firefighters and the control. The *X* axis shows the centered log transformed values of the original counts while the *Y* axis displays their density. The horizontal box plots show the distribution and the respective median values for each group. (**a**) The 2-Dehydro-3-deoxyphosphogluconate aldolase (EDA), (**b**) Dipeptidases Kos, (**c**) Sortase A, (**d**) DNA segregation ATPase FtsK, (**e**) Glycoside Hydrolase Family 25, (**f**) Glycosyl Transferase Family 51, (**g**) Carbohydrate-Binding Module Family 13, and (**h**) Glycosyl Transferase Family 35.

**Table 1 microorganisms-13-00680-t001:** Demographic Characteristics in firefighters’ groups and the control groups (non-firefighters).

Characteristics	Firefighter		Non-Firefighter	
	Korean	American	Korean	American
Age (Years)	Number (%)			
21–30	13 (26%)	11 (21.2%)	10 (19.6%)	16 (32%)
31–40	27 (54%)	24 (46.2%)	23 (45.1%)	18 (36%)
41–50	10 (20%)	17 (32.7%)	18 (35.3%)	16 (32%)
Total	50 (100%)	52 (100%)	51 (100%)	50 (100%)
Marital Status				
Married	37 (74%)	40 (76.9%)	26 (51%)	25 (50%)
Divorced or separated	0 (0%)	3 (5.8%)	0 (0%)	1 (2%)
Widowed	0 (0%)	1 (1.9%)	0 (0%)	0 (0%)
Never Married	13 (26%)	8 (15.4%)	25 (49%)	24 (48%)
Total	50 (100%)	52 (100%)	51 (100%)	50 (100%)
Education				
High School Diploma	6 (12%)	23 (44.2%)	1 (2%)	6 (12%)
College degree (associates, bachelors)	39 (78%)	27 (51.9%)	39 (76.5%)	22 (44%)
Professional degree (masters, PhD)	5 (10%)	2 (3.8%)	11 (21.6%)	22 (44%)
Total	50 (100%)	52 (100%)	51 (100%)	50 (100%)
Race				
White	0 (0%)	48 (92.3%)	0 (0%)	33 (66%)
Asian	50 (100%)	0 (0%)	51 (100%)	11 (22%)
Black/African American	0 (0%)	2 (3.85%)	0 (0%)	3 (6%)
Other	0 (0%)	2 (3.85%)	0 (0%)	3 (6%)
Total	50 (100%)	52 (100%)	51 (100%)	50 (100%)
Occupational Positions				
Firefighter operators	27 (54%)	22 (42.3%)	--	--
Firefighter administrators	4 (8%)	0 (0%)	--	--
Captain or Chief Officer of firefighters	13 (26%)	12 (23.1%)	--	--
Driver engineer	6 (12%)	11 (21.1%)	--	--
Lieutenant	0 (0%)	7 (13.5%)	--	--
Non-firefighter	0 (0%)	0 (0%)	51 (100%)	50 (100%)
Total	50 (100%)	52 (100%)	51 (100%)	50 (100%)
Job Type				
Firefighter	50 (100%)	52 (100%)	0 (0%)	0 (0.0%)
Healthcare (physiotherapists and caregiver)	--	--	7 (13.7%)	2 (4%)
Researcher	--	--	1 (2%)	12 (24%)
Education	--	--	13 (25.5%)	8 (16%)
Student	--	--	1 (2%)	7 (14%)
Financial Management and Organizer (Operations Director, Community organizer)	--	--	6 (11.8%)	6 (12%)
Sports Performance and Training (athletics and fitness trainers)	--	--	1 (2%)	2 (4%)
Engineering	--	--	0 (0%)	5 (10%)
Food industry	--	--	1 (2%)	1 (2%)
Construction	--	--	1 (2%)	2 (4%)
Running own business	--	--	7 (13.7%)	0 (0%)
Sales	--	--	3 (5.8%)	0 (0%)
Unemployment	--	--	3 (5.8%)	0 (0%)
Unknown	--	--	7 (13.7%)	5 (10%)
Total	50 (100%)	52 (100%)	51 (100%)	50 (100%)
Years of Service in the Fire Department				
5–10 years	32 (64%)	21 (40.4%)	--	--
More than 10 years	17 (34%)	19 (36.5%)	--	--
More than 20 years	1 (2%)	12 (23.1%)	--	--
Total	50 (100%)	52 (100%)		
Consuming Alcohol Beverages				
No	21 (42%)	8 (15.4%)	30 (58.8%)	9 (18%)
Yes	29 (58%)	44 (84.6%)	21 (41.2%)	41 (82%)
Total	50 (100%)	52 (100%)	51 (100%)	50 (100%)
NSAID or other Painkillers Usage				
Yes	2 (4%)	26 (50%)	0 (%)	15 (30%)
No	48 (96%)	26 (50%)	51 (100%)	35 (70%)
Total	50 (100%)	52 (100%)	51 (100%)	50 (100%)

**Table 2 microorganisms-13-00680-t002:** Logistic regression analysis for the incidence of depression.

B		S.E.	Wald	df	Sig.	Exp(B)	95% C.I. for Exp(B)	
							Lower	Upper
Region_ref: America	0.194	0.287	0.455	1	0.5	1.214	0.691	2.132
Occupation_ref: Control	0.713	0.288	6.144	1	0.013	2.04	1.161	3.586
Constant	−0.69	0.256	7.271	1	0.007	0.502		

## Data Availability

The raw sequence data reported in this paper have been deposited in the Genome Sequence Archive (Genomics, Proteomics & Bioinformatics 2021) [70] in the National Genomics Data Center (Nucleic Acids Res 2022) [71], China National Center for Bioinformation/Beijing Institute of Genomics, Chinese Academy of Sciences (GSA: CRA020017), that are publicly accessible at https://ngdc.cncb.ac.cn/gsa/s/0BUBu61b (accessed on 17 March 2025).

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
