# Peer review of "Gut Microbiome Alterations, Mental Health, and Alcohol Consumption: Investigating the Gut–Brain Axis in Firefighters"

_microorganisms, 2025, doi:10.3390/microorganisms13030680_

Round 1
Reviewer 1 Report
Comments and Suggestions for Authors
Congratulations to authors,
this is very interesting paper, with significant contribution to overall pathogenesis of the gut-brain and stress-related disorders among susceptible population like firefighters in two different world regions.
The abstract however, needs to be slightly improved. Please add aims.
Ln 28 delete 'after excluding......'
Ln 42 reducing the loss of life? Please rewrite.
Ln 49-50 No need to write 0 after . in case of % (f. example 52% instead of 52.0% etc)
These observations have led to a possible role of the 64
Ln 64 I didn't understand the sentence: ....possible role of gut microbiota in alcohol consumption.
Ln 137 If a participant.....instead of when a participant
Ln 321 please remove 'b.'
Table 1. Please improve formatting.
Also, I didn't understand why did you choose to correlate PTSD and gut microbiota, and not anxiety and depression? Also, I didn't quite get the % of respondents who had anxiety, depression, and PTSD and combined disorders/symptoms?
Insteresting would be if you'd correlate (for the next research) the levels of /the expression of enzymes responsible for alcohol metabolism among respondents (alcohol dehydrogenase, aldehyde dehydrogenase) among firefighters and general population
The discussion is greatly written, very clear and understanding.
Please reconsider going through the entire manuscript carefully one more time for potential english language improvement.
Comments on the Quality of English Language
The English could be improved to more clearly express the research.
Reviewer 2 Report
Comments and Suggestions for Authors
The manuscript by Ji Youn Yoo et al. explored the gut-brain axis in firefighters. The study is interesting. I have the following questions and comments:
1, the details of the statistical analysis must be provided. Please revise the Materials and Methods section accordingly.
2, in figure 1b, I think it should be "American" and "Korean", not "English" and "Korean".
3, the statistics in Figures 4 and 5 are missing. Please revise.
4, the limitations of the study must be further discussed.
5, why drinking could change the abundance of Bifidobacterium bifidum? Any explanation for that?
Reviewer 3 Report
Comments and Suggestions for Authors
The paper deals with a fascinating topic, exploring a very hot topic, that of the gut-brain axis.
The paper's pros include showing a study in which two populations from different geographical areas were considered, clearly defining the aim and the endpoints, and clearly illustrating the methods used for data collection, strengthening the robustness of the design.
The main con is that of having a small sample size.
Here are some suggestions:
- Check the relative frequency percentages in Table 1
- In general, the interpretation of the data must be written with more scientific rigor, reporting the p values ​​in the text to confirm the conclusions that the authors reach by stating the results (such as lines 219-220 on page 7)
- It could be helpful to report the p values ​​in all the figures to make them immediately assessable or use *,**,*** for p values ​​​​respectively ≤ 0.05, ≤ 0.01, ≤ 0.001
- Clarify what type of logistic regression model was applied (univariate, multivariate) and, eventually, what was adjusted for. It would be helpful to summarize the significant results in an additional table
- Figs 4a and 4b are not very clear. Specify the difference not only in the caption
- Page 15. Lines 339-341: It would be beneficial to include a separate paragraph discussing the study's limitations, particularly regarding the small sample size. Additionally, the observed differences in gut microbiome diversity between American and Korean participants could be attributed to the significant intra- and inter-personal variability of gut microbiota. This variability is influenced by various factors, including geographical origin.
- Page 17, line 439: Argue the importance of the gut-brain axis since, in the last ten years, a strong association has been found between changes in the composition of the microbiota and various pathological conditions of the host. In particular, the intestinal microbiome has emerged as a target organ that influences the development of some diseases. (doi: 10.1007/s00592-023-02088-x )
- Page 17, line 447: Discuss more about Alistipes. Alistipes are anaerobic Gram-negative bacteria of the phylum Bacteroidetes commonly present in the healthy human gut microbiota (doi: 10.14336/AD.2023.0118). It was reported that Japanese centenarians showed a pronounced abundance of fecal Alistipes compared to younger controls; the authors proposed this finding as a potential marker of successful aging (doi: 10.1038/s41586-021-03832-5.).
Round 2
Reviewer 2 Report
Comments and Suggestions for Authors
The authors have revised the manuscript accordingly. It can be considered for publication.
Reviewer 3 Report
Comments and Suggestions for Authors
The authors have modified the manuscript, improving its value